# Hepatitis B virus seroepidemiology data for Africa: Modelling intervention strategies based on a systematic review and meta-analysis

Anna L. McNaughton[1], José Lourenço[2], Phillip Armand Bester[3], Jolynne Mokaya[1], Sheila F. Lumley[1,4], Uri Obolski[5,6], Donall Forde[7], Tongai G. Maponga[8], Kenneth R. Katumba[9], Dominique Goedhals[3], Sunetra Gupta[2], Janet Seeley[9,10], Robert Newton[9,11], Ponsiano Ocama[12], Philippa C. Matthews[1,4]*

1 Nuffield Department of Medicine, University of Oxford, Medawar Building for Pathogen Research, Oxford, United Kingdom, 2 Department of Zoology, University of Oxford, Medawar Building for Pathogen Research, Oxford, United Kingdom, 3 Division of Virology, University of the Free State and National Health Laboratory Service, Bloemfontein, South Africa, 4 Department of Infectious Diseases and Microbiology, Oxford University Hospitals NHS Foundation Trust, John Radcliffe Hospital, Headington, Oxford, United Kingdom, 5 School of Public Health, Tel Aviv University, Tel Aviv, Israel, 6 Porter School of the Environment and Earth Sciences, Tel Aviv University, Tel Aviv, Israel, 7 Nuffield Department of Medicine, Nuffield Department of Medicine Research Building, Headington, Oxford, United Kingdom, 8 Division of Medical Virology, University of Stellenbosch, Faculty of Medicine and Health Sciences, Cape Town, South Africa, 9 Medical Research Council/Uganda Virus Research Institute and London School of Hygiene and Tropical Medicine Uganda Research Unit, Entebbe, Uganda, 10 Faculty of Global Health and Development, London School of Hygiene and Tropical Medicine, London, United Kingdom, 11 Department of Health Sciences, University of York, York, United Kingdom, 12 Makerere University College of Health Sciences, Kampala, Uganda

* philippa.matthews@ndm.ox.ac.uk

## Abstract

### Background

International Sustainable Development Goals (SDGs) for elimination of hepatitis B virus (HBV) infection set ambitious targets for 2030. In African populations, infant immunisation has been fundamental to reducing incident infections in children, but overall population prevalence of chronic hepatitis B (CHB) infection remains high. In high-prevalence populations, adult catch-up vaccination has sometimes been deployed, but an alternative Test and Treat (T&T) approach could be used as an intervention to interrupt transmission. Universal T&T has not been previously evaluated as a population intervention for HBV infection, despite high-profile data supporting its success with human immunodeficiency virus (HIV).

### Methods and findings

We set out to investigate the relationship between prevalence of HBV infection and exposure in Africa, undertaking a systematic literature review in November 2019. We identified published seroepidemiology data representing the period 1995–2019 from PubMed and Web of Science, including studies of adults that reported prevalence of both hepatitis B

**Data Availability Statement:** Data from the systematic literature review to determine the

relationship between HBsAg (prevalence of HBV infection) and anti-HBc (exposure to HBV infection) in African populations is available on Figshare via the following link: https://figshare.com/s/ 4414fce1d474bc8a6198. Visualisation of HBV sero-epidemiology data for Africa is available on Shinyapps (RStudio) via the following link: https:// hbv-geo.shinyapps.io/oxafricahbv/. Please note that links to the metadata have been appropriately referenced throughout the manuscript.

**Funding:** PCM is funded by an intermediate fellowship from the Wellcome Trust, which supported this work (grant reference number 110110). The funders had no role in study design, data collection and analysis, decision to publish, or preparation of the manuscript.

**Competing interests:** The authors have declared that no competing interests exist.

**Abbreviations:** anti-HBc, antibody to hepatitis B core antigen; anti-HBs, antibody to hepatitis B surface antigen; ART, antiretroviral therapy; cccDNA, covalently closed circular DNA; CHB, chronic hepatitis B; EPI, Expanded Programme for Immunization; HBeAg, hepatitis B e-antigen; HBIg, hepatitis B immunoglobulin; HBsAg, hepatitis B surface antigen; HBV, hepatitis B virus; HCC, hepatocellular carcinoma; HCV, hepatitis C virus; HDV, hepatitis delta virus; HIV, human immunodeficiency virus; HLA, human leukocyte antigen; LRT, likelihood ratio test; MCMC, Markov chain Monte Carlo; NA, nucleoside/nucleotide analogue; PMTCT, prevention of mother to child transmission; PRISMA, Preferred Reporting Items for Systematic Reviews and Meta-Analyses; SDGs, Sustainable Development Goals; T&T, Test and Treat; T4P, Treatment for Prevention; TDF, tenofovir disoproxil fumarate; UN, United Nations.

surface antigen (HBsAg; prevalence of HBV infection) and antibody to hepatitis B core antigen (anti-HBc; prevalence of HBV exposure). We identified 96 studies representing 39 African countries, with a median cohort size of 370 participants and a median participant age of 34 years. Using weighted linear regression analysis, we found a strong relationship between the prevalence of infection (HBsAg) and exposure (anti-HBc) ($R^2 = 0.45$, $p < 0.001$). Region-specific differences were present, with estimated CHB prevalence in Northern Africa typically 30% to 40% lower ($p = 0.007$) than in Southern Africa for statistically similar exposure rates, demonstrating the need for intervention strategies to be tailored to individual settings. We applied a previously published mathematical model to investigate the effect of interventions in a high-prevalence setting. The most marked and sustained impact was projected with a T&T strategy, with a predicted reduction of 33% prevalence by 20 years (95% CI 30%–37%) and 62% at 50 years (95% CI 57%–68%), followed by routine neonatal vaccination and prevention of mother to child transmission (PMTCT; at 100% coverage). In contrast, the impact of catch-up vaccination in adults had a negligible and transient effect on population prevalence. The study is constrained by gaps in the published data, such that we could not model the impact of antiviral therapy based on stratification by specific clinical criteria and our model framework does not include explicit age-specific or risk-group assumptions regarding force of transmission.

## Conclusions

The unique data set collected in this study highlights how regional epidemiology data for HBV can provide insights into patterns of transmission, and it provides an evidence base for future quantitative research into the most effective local interventions. In combination with robust neonatal immunisation programmes, ongoing PMTCT efforts, and the vaccination of high-risk groups, diagnosing and treating HBV infection is likely to be of most impact in driving advances towards elimination targets at a population level.

## Author summary

### Why was this study done?

- Hepatitis B virus (HBV) infection is a major global health problem, with an estimated 290 million infections worldwide; international targets set the challenge for this public health threat to be eliminated by 2030.

- Administering HBV vaccine to babies is a very successful way to prevent new infections, but previous studies have shown that this strategy will take many decades to bring about elimination targets.

- We set out to investigate other approaches that can be used in combination with the infant vaccination schedule, using data from a meta-analysis and modelling the impact of vaccinating older children and adults, or enhancing testing and treatment (T&T) of existing HBV infections.

## What did the researchers do and find?

- We undertook a review and meta-analysis of all the published data describing the frequency of HBV infection, as well as the frequency of exposure to HBV infection in African populations, working with data from 96 studies published between 1995 and 2019.

- Using these data, we demonstrated a significant relationship between exposure and infection (overall $p < 0.001$) and identified differences between geographic regions.

- Using an existing mathematical model, our findings suggest that vaccinating older age groups has negligible sustained effect on HBV rates in a population, but universal T&T of HBV is predicted to have a substantial impact.

## What do these findings mean?

- Our results show different patterns of HBV infection and transmission in different regions of Africa, illustrating that interventions may need to be tailored according to the specific setting.

- We have demonstrated that vaccination campaigns targeting older children and adults are unlikely to be an effective use of resources in most African populations, and that resources are better diverted to improving diagnosis and treatment of existing infections.

- The study is limited by gaps in existing data, such that many populations in Africa are poorly represented. Our conclusions are drawn from the output of a model and will need to be validated in clinical practice.

## Introduction

There is an estimated global burden of 290 million cases of hepatitis B virus (HBV) infection [1], the majority of which are undiagnosed and untreated [2]. Prevalence of HBV exposure and chronic hepatitis B (CHB) infection are extremely high in some settings in Africa [3, 4]. Robust epidemiology data are lacking, and some populations in Africa have specific vulnerabilities associated with poverty, stigma, and co-endemic human immunodeficiency virus (HIV) infection [2]. Horizontal transmission within households, particularly affecting young children, is a significant acquisition route in some African populations [5, 6], but the specific mechanism and timing of transmission remain uncertain in many cases.

Vaccination to protect against infection is a cornerstone of interventions, with enhanced efforts arising as a result of United Nations (UN) Sustainable Development Goals (SDGs) setting out elimination targets for the year 2030 [7]. Vaccination is included in the Expanded Programme for Immunization (EPI) and has been progressively rolled out for infants across Africa since 1995, alongside enhanced interventions for the prevention of mother to child transmission (PMTCT) [8]. However, despite 2 decades of vaccine implementation, CHB remains endemic in many regions, and published mathematical models exploring the impact

of existing interventions demonstrate time scales for success that are substantially beyond the 2030 targets [9, 10].

Treatment of CHB is typically based on nucleoside/nucleotide analogue (NA) drugs, among which tenofovir disoproxil fumarate (TDF) is the most affordable, accessible, and widely recommended [8, 11–13]. NA agents have a high rate of success in suppressing detectable viraemia, but do not tackle the liver reservoir of virus (in the form of covalently closed circular DNA (cccDNA) in the nucleus of infected hepatocytes [14]), so most individuals are committed to long-term treatment. Individuals with CHB are assessed for treatment based on algorithms that stratify the long-term risk of liver disease based on age and sex, laboratory measurement of liver enzymes and HBV DNA viral load, elastography scores, or other imaging data [11–13]. Treatment guidelines also recognise and support the use of TDF during pregnancy as part of an approach to reduce vertical transmission [13], and this approach is widely accepted (albeit variably deployed) [15]. However, there is less recognition of the wider potential for NA agents to be deployed as a public health intervention to reduce new incident cases.

Tackling the large, chronic population reservoir of HBV in adults is important, and various strategies are employed to reduce new incident infections in adults. There is an interesting conflict between formal recommendations and public health policies that are deployed in practice. Although evidence suggests that 'catch-up' vaccination campaigns are likely to have limited population benefit [9] and are not endorsed by guidelines [8], the strategy can appear an attractive public health approach for older children and adults in high-prevalence populations [16] and has been undertaken in some settings [17]. Economic analyses have reported that catch-up campaigns in young adults are cost-effective only if combined with screening [18], highlighting the importance of focusing not only on prevention but also on investment in diagnosis and treatment [19]. The latter concept has been embraced for HIV under the banners of Treatment for Prevention (T4P) and, more recently, Universal Test and Treat (T&T) [20], recognising that antiretroviral treatment (ART) confers benefits both to the individual and also to populations by reducing the risk of transmission [21, 22].

Building on this experience from HIV, a T&T HBV strategy could offer substantial advantages. The feasibility, acceptability, and public health consequences of this approach have been discussed in reports of HBV treatment deployment in The Gambia [19, 23], although previous approaches treat only selected individuals deemed to be at highest risk. Evidence-based consideration of the impact of population-level interventions, including more inclusive approaches for treatment of a greater proportion of all individuals with HBV, are urgently required if we are to accelerate progress towards elimination targets.

We used a systematic literature review to investigate the seroepidemiology of HBV across the African subcontinent in order to highlight regional differences and inform local strategies, with a specific focus on interventions that target adults. Recognising that catch-up campaigns are already being deployed in some locations, we considered the evidence for any benefit of this approach, and in parallel, we assessed the impact of universal T&T using epidemiological data and a mathematical model [9].

## Methods

### HBV seroepidemiology for Africa

We undertook a systematic search of PubMed and Web of Science in November 2019, using Preferred Reporting Items for Systematic Reviews and Meta-Analyses (PRISMA) criteria to underpin a predefined protocol for data collection (Fig 1; S1 PRISMA Checklist). We used the search terms 'HBV antibody,' 'anti-HBc,' 'HB core antibody,' 'HBV exposure,' or 'HBV prevalence' AND 'Africa' or [Name of specific country], using the UN geoscheme for Africa to

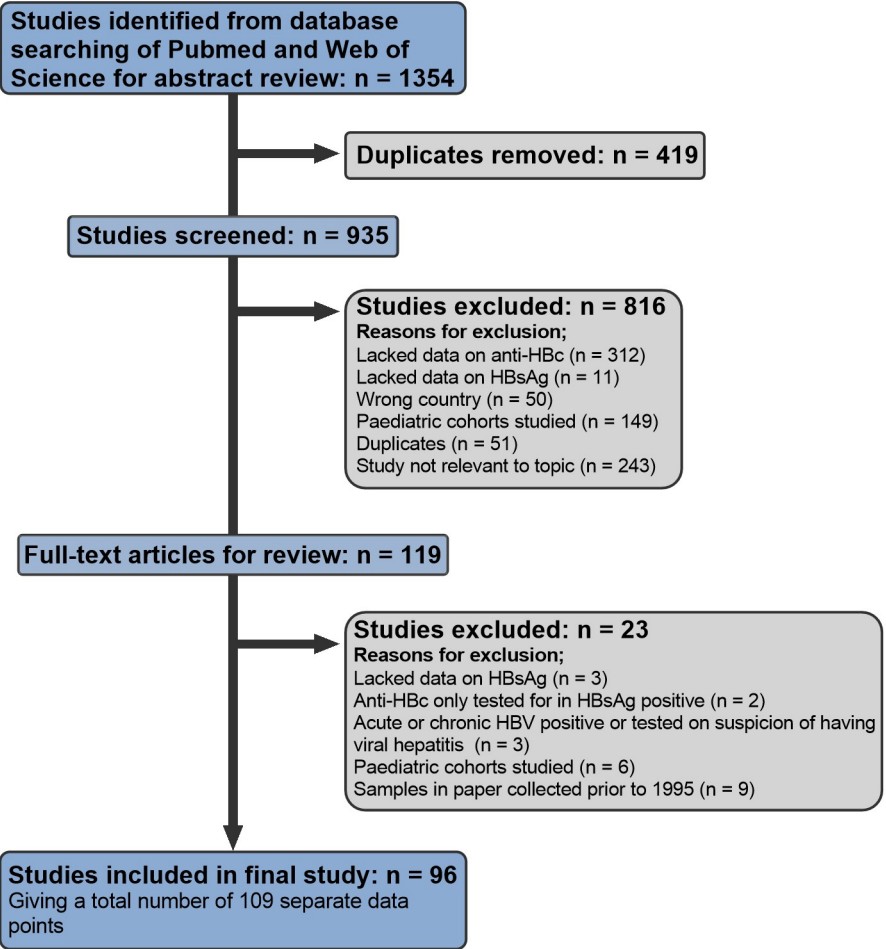

**Fig 1. PRISMA flowchart showing selection of studies of HBsAg and anti-HBc prevalence in Africa, published 1995–2019, for inclusion in meta-analysis.** The analysis included 96 studies; 13 of these studies reported data for 2 discrete cohorts, giving a total of 109 data points included in the final analysis. Anti-HBc, antibody to hepatitis B core antigen; HBsAg, hepatitis B surface antigen; HBV, hepatitis B virus; PRISMA, Preferred Reporting Items for Systematic Reviews and Meta-Analyses.

define a list of countries [24]. We assessed the quality of included studies using recommended criteria from the Joanna Briggs Institute [25, 26]. Inclusion criteria were as follows:

- Data gathered after widespread roll-out of infant vaccination in 1995

- No reported data collection undertaken pre 1995

- Reported prevalence of both hepatitis B surface antigen (HBsAg) and antibody to hepatitis B core antigen (anti-HBc)

- Cohorts primarily reporting data for adults (age $\geq$ 16 years)

- Cohort does not sample a population enriched for HBV infection (specific exclusions are presented in Fig 1)

We recorded anti-HBc prevalence (i.e., proportion of population exposed to HBV, irrespective of chronic infection status, also termed 'total exposure') and also calculated the proportion

of the population with cleared infection (i.e., anti-HBc prevalence minus HBsAg prevalence, termed 'exposed and cleared'). For studies reporting prevalence data from ≥2 cohorts (e.g., HIV-positive and HIV-negative populations), we recorded these as a single publication but ≥2 discrete data points. We also sought evidence for recommendations underpinning catch-up vaccination of adolescents and adults in Africa cited in PubMed using the search terms 'hepatitis b virus' or 'HBV,' and 'Africa' or [individual country name], with 'vaccin*' and 'catch up' or 'adult.' All studies were evaluated independently by 2 or more authors prior to inclusion in the analysis.

We set out with 2 prespecified aims, (i) to investigate the relationship between the prevalence of CHB infection and HBV exposure for different regions of Africa and (ii) to use an existing model to predict the outcomes of population interventions in a high-prevalence setting.

### Statistical analysis of metadata

In order to derive an estimation of the age group represented by the studies included, we used mean (reported by 48/109 cohorts) and median (reported by 23/109 cohorts). For the remaining 38 cohorts, age was not reported, or the data could not be disaggregated.

The UN geoscheme includes 5 African regions: Central, Eastern, Northern, Southern, and Western [24]. We analysed prevalence data for anti-HBc and HBsAg using Graphpad Prism version 7.0 and R. We used weighted linear regression based on cohort size to derive lines of best fit and 95% confidence intervals and to interpolate data points. We generated maps using R [27, 28].

Estimates of pooled median proportions (HBsAg positive, anti-HBc), were obtained through logistic regression with a random intercept assigned to each cohort, performed separately for each African region. The 95% confidence intervals of the proportions were based on profile likelihood of the model's fixed effect intercepts. Comparisons between different regions were obtained by sub-setting the data for each pair of regions and creating 2 models. The first was the logistic regression model with a random intercept assigned to each cohort described earlier, while the second had an added fixed effect of the study region. These 2 models were compared through the likelihood ratio test (LRT). The obtained p-values from the LRTs were then adjusted for multiple comparisons using the Holm-Bonferroni method, and the adjusted p-values are reported.

### Modelling the impact of adult vaccination versus T&T

In this study, we adapted a previously published HBV dynamic model and Markov chain Monte Carlo (MCMC) approach that we initially developed to fit the seroepidemiology of Kimberley in South Africa to project the impact of interventions in that transmission setting [9]; as our mathematical and computational methods are already published, we have not replicated them in this paper. However, for ease of reference, we have provided a summary overview of model population classes in Fig 2, as well as relevant parameters in S1 Table.

In this instance, we fitted the model to data from Uganda, a setting of high HBsAg prevalence [3]. Target (model-fitted) seroepidemiology variables were HBsAg prevalence at approximately 10%, anti-HBc prevalence at 42%, and hepatitis B e-antigen (HBeAg) positive relative prevalence at 27% (S1 Table). We left 4 model parameters free to be fitted to the Uganda setting (with uninformative priors): vertical transmission rate for HBeAg-positive and HBeAg-negative groups, rate of conversion from HBeAg positive to HBeAg negative, and spontaneous clearance of chronic HBV infection. The MCMC fitted the model closely to the target variables at 10% (95% CI 7.92%–11.7%) for HBsAg prevalence, 42.1% (95% CI 40.2%–44.0%) for anti-

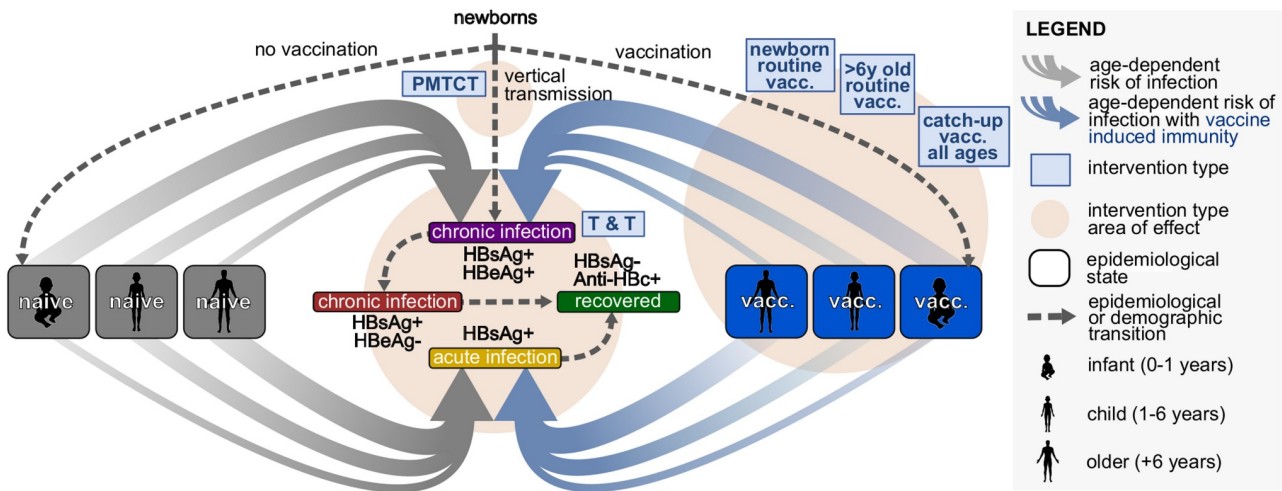

**Fig 2. Summary diagram of the dynamic model used to fit the seroepidemiology of HBV in Uganda.** The naïve (grey rounded boxes) and vaccinated (blue rounded boxes) subpopulations are divided into 3 relevant age groups: infant (0–1 years), child (1–6 years), and older (+6 years). This allows us to consider a relationship between the likelihood of developing chronic or acute infection with age (represented by the thickness of the grey and blue arrows). Individuals can acquire infection in any age group. When acutely infected (yellow rounded box), individuals may clear infection and recover (green rounded box). When chronically infected, individuals can be HBeAg positive (purple rounded box) and transit to become HBeAg negative (red rounded box), after which they may recover (green rounded box). Interventions (light blue squared boxes) target different epidemiological states and transitions (orange circles). Vertical transmission can occur from infected mothers to newborns. Protection mediated by infant vaccination is age dependent due to waning and is assumed to lower susceptibility to infection. Parameters are listed in S1 Table. Artwork supported by figures created with BioRender.com. anti-HBc, antibody to hepatitis B core antigen; HBeAg, hepatitis B e-antigen; HBsAg, hepatitis B surface antigen; HBV, hepatitis B virus; PMTCT, prevention of mother to child transmission; T&T, Test and Treat.

HBc prevalence, and 26.9% (95% CI 24.8%–29.0%) for HBeAg-positive relative prevalence. The posteriors of the free parameters also closely matched literature expectations (S1 Table).

PMTCT and vaccine-based interventions were modelled as previously described (Fig 2) [9]. As the model framework does not allow for risk stratification beyond age groups, we included a simple T&T intervention that reduces the transmission potential of the HBV infected proportionally to control effort (e.g., 20% T&T coverage equated to a 20% reduction in the force of infection). The universal T&T approach thus assumes that HBsAg-positive cases were treated irrespective of any specific risk stratification.

## Results

### Studies identified through a systematic literature review

We collated and curated a total of 96 studies spanning 39 African countries and generating 109 unique data points representing independent cohorts (Fig 1; full metadata online at Figshare: 10.6084/m9.figshare.6154598). Studies reporting prevalence data from ≥2 cohorts (*n* = 13) are summarised in S2 Table. Across all 109 cohorts, the median number of individuals sampled was 370 (IQR 177–638, range 50–9,486). The median ages for the cohorts represented was 34 years (IQR 29–38 years) based on age data reported for 65% of cohorts.

### Quality appraisal of data identified through systematic literature review

We undertook a quality assessment of all the studies we identified (full quality appraisal data online at Figshare: 10.6084/m9.figshare.6154598). The majority of studies included in the meta-analysis were of good quality, with 85% of studies reporting clear details of the methods used to measure HBsAg and anti-HBc reflecting our stringent criteria for inclusion of these

prevalence values. Among 8 of the remaining studies with the poorest quality evaluations, we were unable to evaluate quality of laboratory methods in 7 cases because the journals in question did not support electronic archives dating back far enough to access the full publication, but relevant serological parameters were available from the abstract.

The quality assessment (online at Figshare: 10.6084/m9.figshare.6154598) and Forest Plots of the data (S1 Fig) highlighted differences in our confidence in assessment of prevalence of HBsAg and anti-HBc, with sufficient sample sizes reported by 83% and 53% of all included studies for these 2 parameters, respectively. In almost all cases ($n = 108$), HBsAg and anti-HBc were measured in the same cohorts, and the anti-HBc prevalence was frequently considerably higher than the HBsAg, generally requiring larger cohort sizes to ascertain similar levels of confidence. A number of studies reported powering their analysis for assessing HBsAg prevalence, but this was infrequently considered for anti-HBc.

## Prevalence of infection (HBsAg) is positively correlated with exposure (anti-HBc)

Cohort spatial distribution and associated serological markers are shown in Fig 3 and can be visualised using our interactive tool online [29]. Pooling data for all regions, the prevalence of CHB was positively correlated with exposure ($R^2 = 0.45$, $p < 0.001$ by weighted linear regression) (Fig 4A). Prevalence of CHB differs significantly between regions (Fig 4B–4F). Focusing specifically on Uganda, we also found a significant relationship between CHB prevalence and exposure ($p = 0.006$, Fig 4G), although considerable CHB prevalence/exposure ratio variation is seen between different Ugandan studies.

There are striking differences in seroprevalence by country (Fig 5A) and by region (Fig 5B). For example, HBsAg prevalence in Northern Africa is lower compared to Western and Southern Africa ($p < 0.001$ and $p = 0.007$, respectively, Fig 5B), but this difference cannot be explained only by lower population exposure (anti-HBc) rates: although exposure is lower in Northern than Western Africa ($p < 0.001$), there is no difference in exposure between Southern and Northern Africa ($p = 0.5$). Indeed, the predicted CHB prevalence was generally 30% to 40% lower in Northern than Southern Africa for any given exposure (Fig 4D and 4E; S3 Table). Central African regions display a different relationship between HBsAg and anti-HBc, compared to other settings whereby high exposure is not associated with a correspondingly high prevalence of CHB infection (Fig 4C). This is likely to be a robust representation of the region, as the data are presented by 14 studies published over a 16-year period and represent multiple countries in which a median of 383 individuals were analysed (IQR 201–772 individuals).

We did not find any significant differences in prevalence or exposure between HIV-positive cohorts ($n = 27$) and all other cohorts ($n = 79$; both $p > 0.05$; S2 Fig). In 3 studies that assessed both HIV-positive and HIV-negative cohorts, CHB prevalence was consistently higher among HIV-positive patients (mean 2.23-fold) [30–32]. Exposure was also higher in HIV-positive cohorts than in HIV-negative cohorts for two-thirds of studies [31, 32]. In a third study in South Africa, exposure was similar irrespective of HIV status, suggesting that the increased CHB prevalence in the HIV-positive cohort was the result of reduced clearance rates relative to the HIV-negative cohort [30].

## Impact of catch-up vaccination of adolescents and adults in highly endemic settings

We did not identify any published evidence or specific recommendations for catch-up of adolescents and adults. However, a number of authors suggest catch-up programmes as a way of

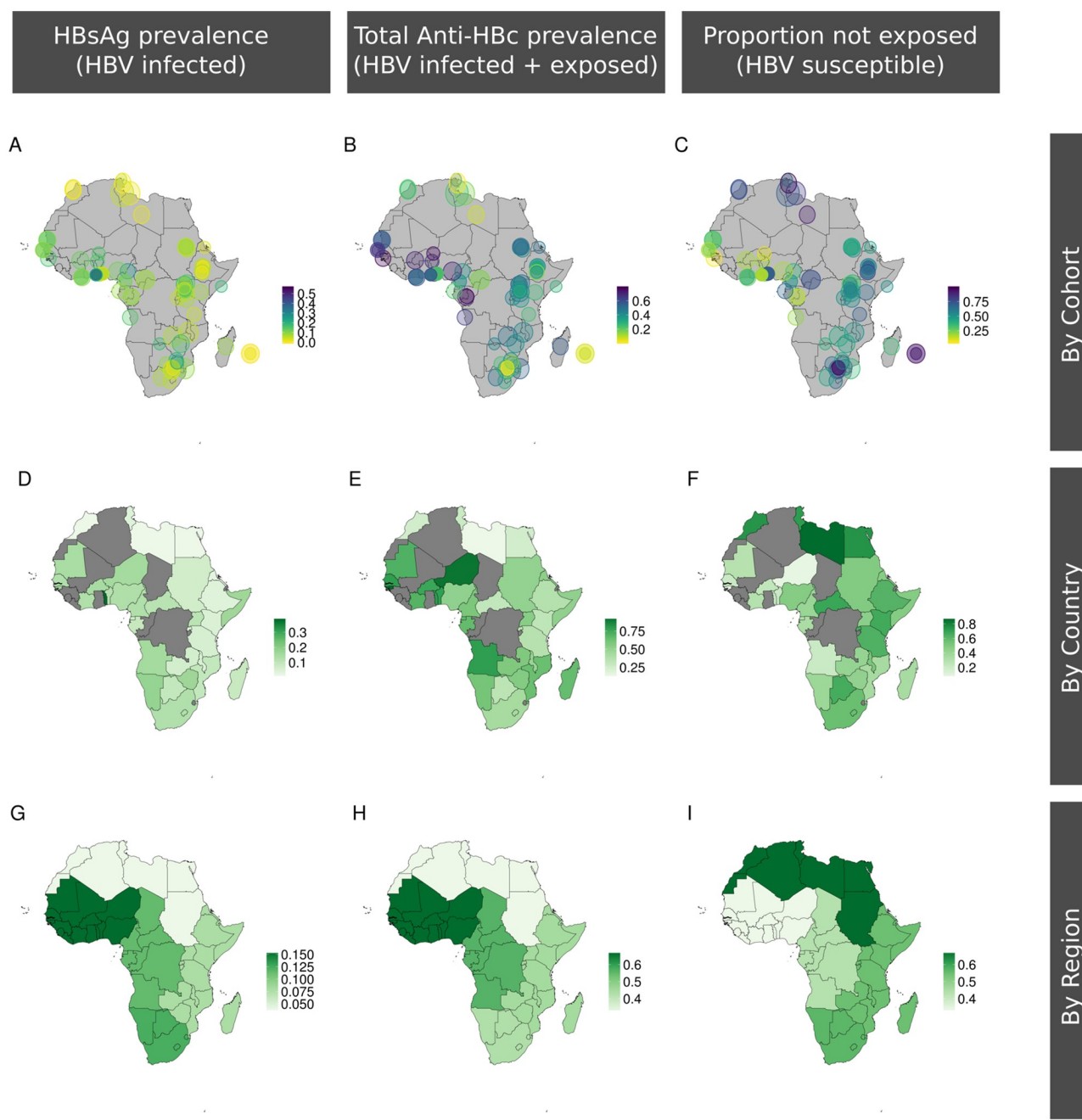

**Fig 3. Maps demonstrating the location and HBV seroepidemiology of adult cohorts identified through a systematic literature review.** First row shows data by individual cohort, depicting (A) HBsAg prevalence, (B) anti-HBc prevalence (exposed plus infected), and (C) HBV susceptible population (100% of population minus anti-HBc prevalence). Each circle is placed to represent the location of the cohort. Second row shows data by country (D–F) and third row by region (G–I). Each area is coloured to reflect high to low prevalence of the attribute in question (scale bar as shown on each panel). Countries shown in grey have no data. The cohort metadata are available online at Figshare: 10.6084/m9.figshare.6154598, and an interactive version of these maps can be accessed online [29]. Outline maps available as open source from http://www.maplibrary.org/library/index. htm. Anti-HBc, antibody to hepatitis B core antigen; HBsAg, hepatitis B surface antigen; HBV, hepatitis B virus.

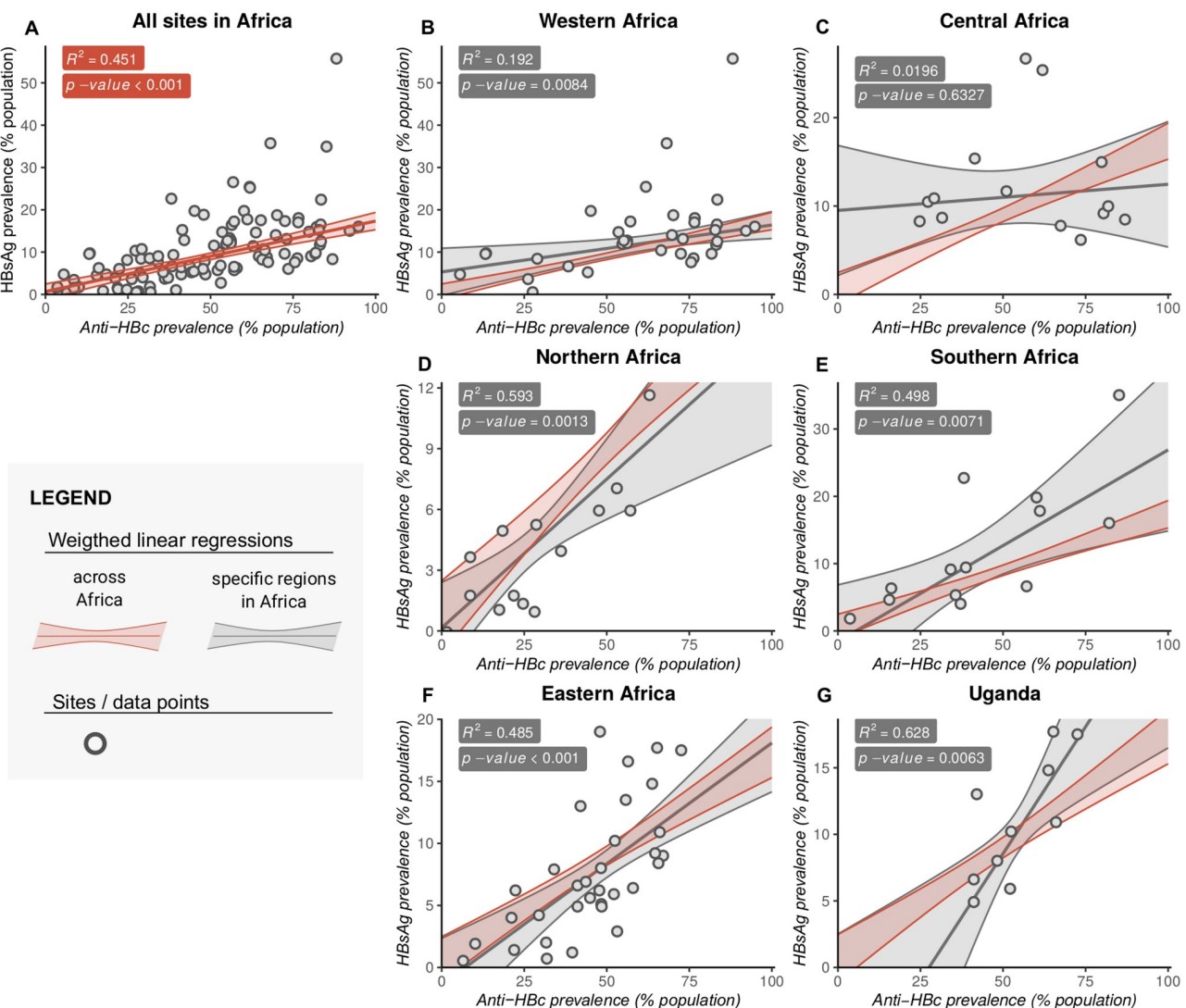

**Fig 4. Relationship between population prevalence of anti-HBc (exposure) and HBsAg (active infection) for different regions of Africa.** Data are shown for (A) the entire African continent; for (B) Western, (C) Central, (D) Northern, (E) Southern, and (F) Eastern Africa; and for (G) Uganda. These data are derived from a review of the published literature, using UN geoschemes to classify the geographic regions [24]. $R^2$ and p-values by weighted linear regression (solid line), with studies weighted by cohort size. Outer dashed lines show 95% confidence intervals. Weighted linear regression plots and 95% CIs (shaded regions) are shown for the whole of Africa in red, for each region in grey. Linear regression lines of fit in plots B–G have been shown together with the fit for the whole continent for comparison. In each plot, the dots represent only studies associated with that region. Anti-HBc, antibody to hepatitis B core antigen; HBsAg, hepatitis B surface antigen; UN, United Nations.

tackling high HBV prevalence [3, 16, 33, 34] (all the relevant studies and recommendations are summarised in S4 Table).

Based on combining the mean prevalence values from Uganda cohorts to provide a broad overview, 54% of adults across this country have been exposed to HBV (among these, 11% of adults are actively HBV infected, and the remainder have serological evidence of resolved infection). This leaves 46% of the total adult population potentially susceptible (orange bars, Fig 5A). Only a small proportion of this susceptible pool of adults would be exposed to HBV infection each year (estimated in another East African study at 3%–4%) [35]. The natural history of infection in otherwise healthy adults suggests that <5% of exposure events lead to

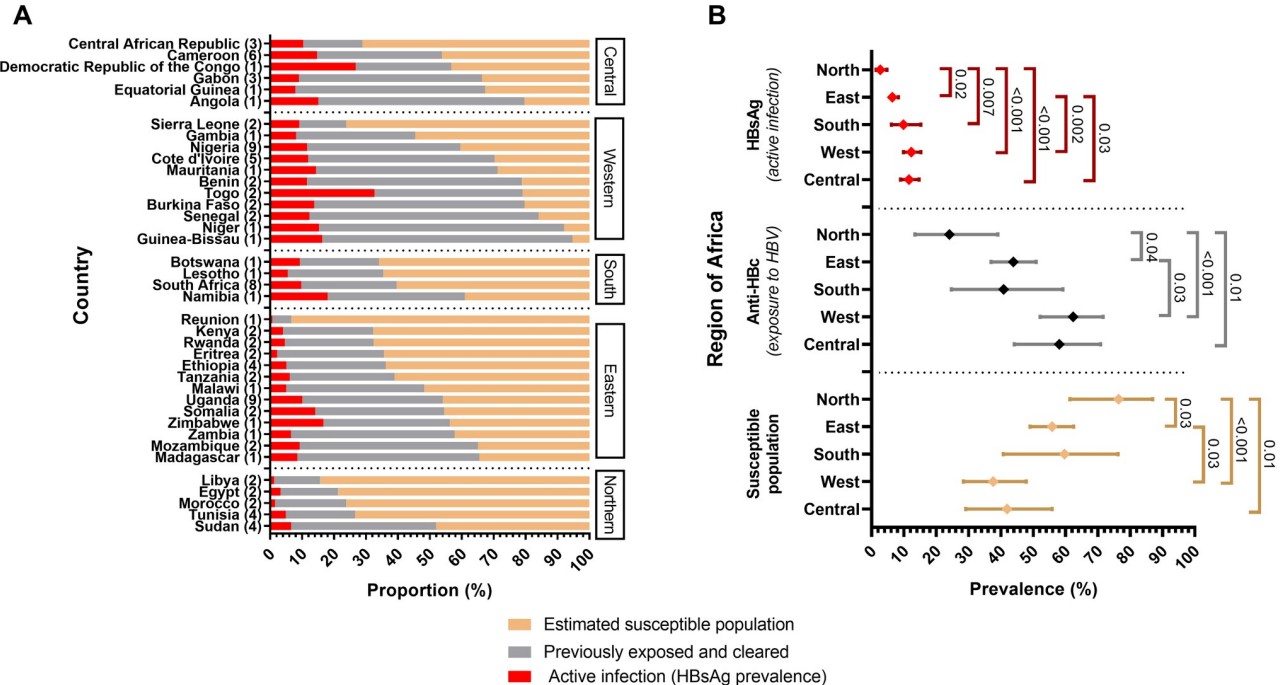

**Fig 5. Estimated proportion of the population with chronic HBV infection (HBsAg), total exposure (anti-HBc), and susceptibility to HBV infection, divided by (A) country and (B) region of Africa.** Countries have been grouped by region according to the UN geoscheme for Africa [24]. The number of studies per country is given in brackets next to the country name. Two studies were counted twice as they contained cohorts from 2 different countries. Per-country estimations were calculated, weighting individual studies according to cohort size. In panel (B), boxplots show the weighted estimations of prevalence with 95% confidence intervals, with significant differences indicated. See methods for definitions of infection, previous exposure, and susceptibility. Anti-HBc, antibody to hepatitis B core antigen; HBsAg, hepatitis B surface antigen; HBV, hepatitis B virus; UN, United Nations.

CHB. Thus, the proportion of the total adult population predicted to avoid CHB through catch-up vaccination each year is, roughly, 50% (vulnerable) × 4% (exposed) × 5% (develop chronicity) = 0.1%.

Using our HBV dynamic model (summarised in Fig 2, and with details previously published [9]), we investigated the impact of catch-up vaccination among adults within a high-prevalence setting, exemplified by Uganda [3] (based on parameters listed in S1 Table). The predicted short-term impact of interventions is shown in Fig 6A–6E with longer-term predictions in Fig 6F. While projecting that catch-up vaccination of all individuals mediates an estimated 16% reduction over 50 years (Fig 6A)—and routine catch-up vaccination of the age group >6 years is predicted to bring about an 11% reduction (Fig 6B)—it is noteworthy that the impact of this strategy disappears over longer time scales (Fig 6F). This decline in impact is due to a limited pool of susceptible adults and the lack of direct impact on the actively infected population. In the long term, this suggests that the strategy offers no sustained overall benefit in progress towards elimination targets, even when deployed at 100% coverage. In contrast, our projections suggest that enhanced coverage of other interventions—including infant immunisation (Fig 6C) and PMTCT (Fig 6D)—could lead to much higher impact, both in the shorter and longer terms.

## Impact of T&T in highly endemic settings

We investigated the projected impact of T&T, concluding that it has the highest impact of the modelled interventions; at 100% coverage, this leads to 33% reduction in prevalence by 20

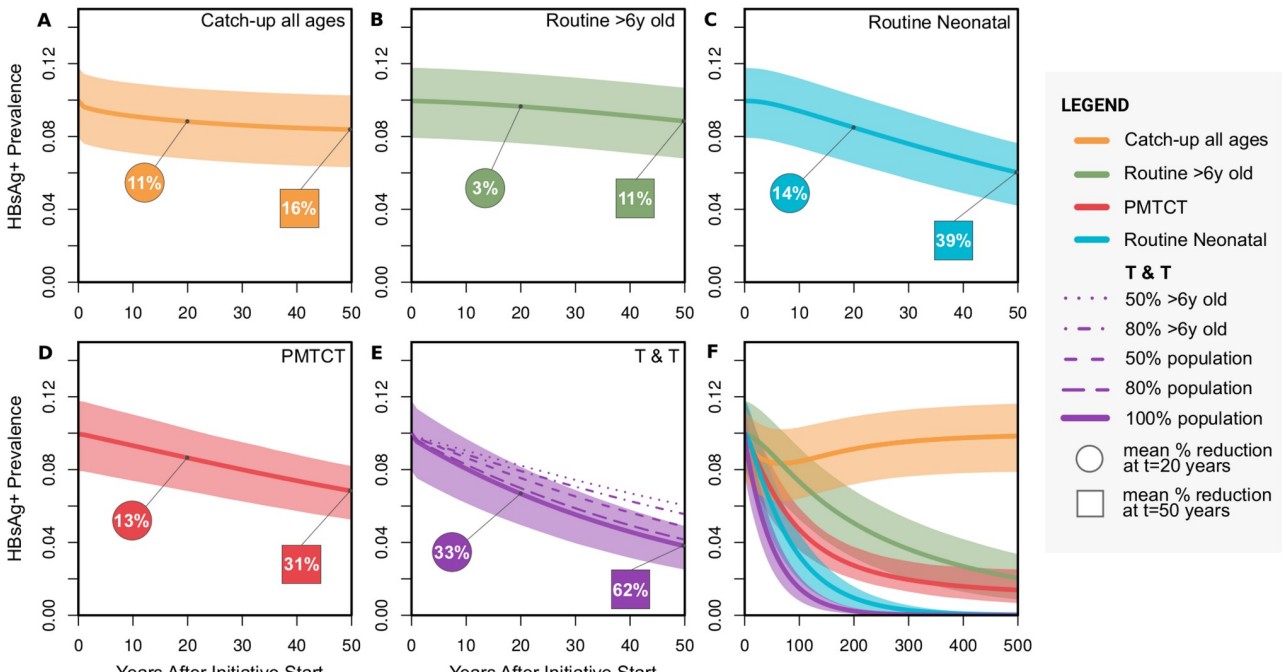

**Fig 6. Decline in HBsAg prevalence over time in response to population interventions for HBV infection in Uganda.** Decline in prevalence is shown over time starting at approximately 10% as used for model fitting to the Uganda setting (see main text). Shaded areas show 95% confidence interval for each intervention based on 5,000 stochastic simulations using parameter samples from the posteriors obtained by fitting the model. Comparisons of interventions applied to 100% of the population: (A) catch-up vaccination of all ages as a one-off event at time = 0 (orange); (B) routine immunisation of children as they reach 6 years of age (proxy for school age) (green); (C) routine neonatal immunisation (blue); and (D) PMTCT interventions for all births (combining accelerated neonatal immunisation with HBIg and antiviral therapy in pregnant mothers, red). (E) Comparison of a T&T approach applied to different proportions of the population: 50% of individuals older than 6 years, 80% of individuals older than 6 years, 50% of population, 80% of population, and 100% of population (purple solid line). This demonstrates the impact of reduced proportions of the population exposed to drug therapy, as a result of either not having drug prescribed or noncompliance with prescribed therapy. (F) Interventions shown in A–E applied to 100% of the population were run for an extended time period of 500 years to observe long-term effects. The numbers at time points t = 20 and t = 50 years, incorporating an overall timeframe that includes the 2030 elimination targets, show the mean reduction in HBsAg prevalence achieved for each of the interventions: (A) 11% (95% CI 9%–15%) and 16% (13%–20%); (B) 3% (2%–5%) and 11% (9%–15%); (C) 14% (12%–18%) and 39% (34%–47%); (D) 13% (10%–15%) and 31% (25%–35%); and (E) 33% (30%–37%) and 62% (57%–68%). HBIg, hepatitis B immunoglobulin; HBsAg, hepatitis B surface antigen; HBV, hepatitis B virus; PMTCT, prevention of mother to child transmission; T&T, Test and Treat.

years (95% CI 30%–37%) and 62% at 50 years (95% CI 57%–68%) (Fig 6E). Recognising the significant barriers to identifying all cases of CHB (which include silent infection, lack of education and awareness, poor access to clinical and laboratory facilities, and stigma [2, 36, 37]), we also modelled the outcome for T&T strategies that reach <100% of the population. Diagnosis and treatment for 80% of infected adults or 50% of the whole infected population (Fig 6E, dashed lines) still deliver a projected reduction in prevalence over time that is comparable to infant immunisation (Fig 6F). Even reducing the target to only 50% of adults (Fig 6E) is still substantially more effective than 100% catch-up vaccination (Fig 6A).

## Discussion

### Summary of main findings

In this study, we set out to evaluate the relationship between the prevalence of CHB infection and exposure in different settings. We suggest that determinants of CHB prevalence involve numerous factors, including age at exposure (Fig 7A), route of infection, host genetics, and viral genotype, and interactions between them remain poorly understood. Using a data set

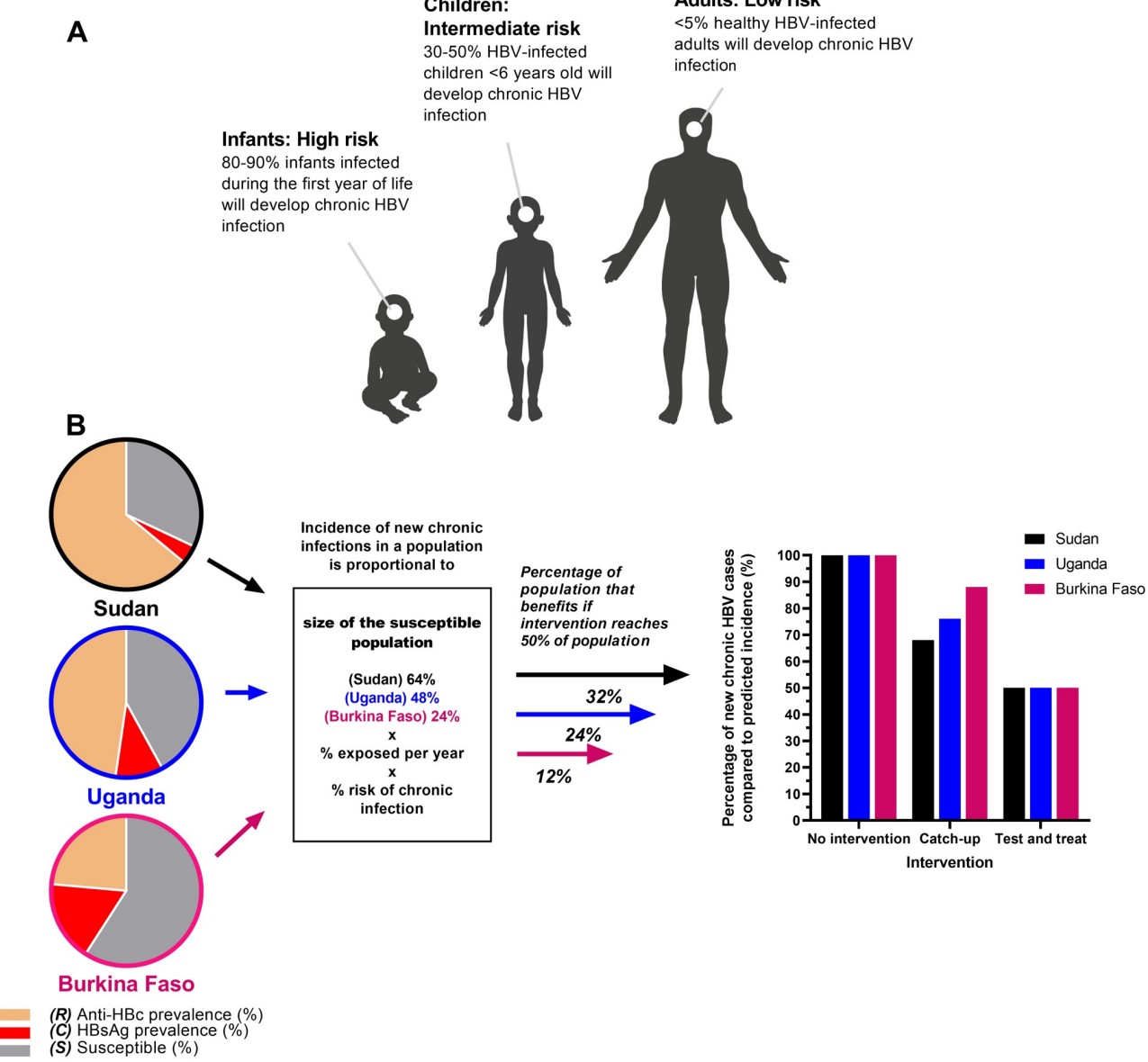

**Fig 7. Cartoons to illustrate seroepidemiology of HBV infection in Africa and the differential impact of HBV interventions according to population targeted.** (A) After exposure to HBV, the risk of developing CHB is highest among young infants, and this risk gradually declines with age until adulthood, when there is low risk of developing CHB. Figure informed by parameters in S1 Table. Artwork supported by figures created with BioRender.com. (B) Populations from Sudan [39], Uganda [3], and Burkina Faso [40] represent approximately the 25th, 50th, and 75th percentiles in the data set collected from our literature review. In adults, assuming that different populations are exposed at the same rate and the risk of CHB is constant (estimated to be 5% in healthy adults as shown in panel A), the incidence of new CHB infection in the population is related to the proportion *S*. Without intervention, 100% of predicted new cases will occur. If 50% of the adult population is vaccinated in a catch-up campaign, CHB will be prevented only among the population *S*. The impact of catch-up vaccination on incidence is therefore related to *S*, with reduced impact in highly exposed populations. In a T&T scenario, with 50% of cases identified and treated, incidence is consistently reduced, regardless of *S*. anti-HBc, antibody to hepatitis B core antigen; *C*, HBsAg prevalence; CHB, chronic hepatitis B; HBsAg, hepatitis B surface antigen; HBV, hepatitis B virus; *R*, anti-HBc prevalence; *S*, susceptible population; T&T, Test and Treat.

gathered via a systematic review, and applying an existing mathematical model, we evaluated the impact of several intervention approaches in a high-prevalence setting (Uganda). Our analysis suggests that catch-up vaccination is of limited impact in this setting and that T&T is predicted to have a much greater, sustained impact that performs most consistently across

settings (Fig 7B). Careful, evidence-based deployment of interventions is essential if sustained and collective progress is to be made towards targets set by the SDGs [7]. Findings of this study highlight the requirement to tailor intervention strategies based on well-informed, high-resolution epidemiological data. This is all the more important for resource-limited settings that are underrepresented in the existing evidence base [38].

## Insights into interventions

The importance of infant vaccination remains paramount and has been well validated in clinical practice, mathematical models, and population epidemiology across diverse settings. In this study, we set out to explore the controversial and less studied approaches of catch-up vaccination and T&T. Although it can seem intuitive to deploy catch-up vaccination for adolescents and adults in high-prevalence settings, only a limited proportion of the population remains susceptible, such that only a minority will benefit from the intervention. Thus, the effectiveness of catch-up is expected to be greater in low-prevalence populations. For this reason, catch-up will frequently not be a prudent use of resources, although in some settings there may be cost benefits in targeting young populations [41], and vaccination may be even more important in regions where hepatitis delta virus (HDV) co-infection is highly prevalent [42].

In contrast, our data suggest that T&T programmes aimed at older children and adults may have great impact, in keeping with the findings of a recent review of vaccination [34] and with economic evaluations [43]. Achieving success through T&T requires multipronged investment including education, laboratory infrastructure, and provision of effective, sustained drug therapy for both HBV infection and HIV/HBV coinfection. Focus on diagnosis is prerequisite [19, 44], parallel investment in infrastructure is required to triage cases for treatment (based on including laboratory and radiological criteria), and scrutiny will be required for drug resistance [45].

The epidemiology and dynamics of infection are different in certain high-risk subgroups (healthcare workers, partners and household contacts of infected individuals, sex workers and their clients, men who have sex with men), and continuing to target these individuals with preventive vaccination remains very important. Likewise, we continue to emphasise the importance of routine infant immunization campaigns, which are a cornerstone of elimination strategies [9].

## Reasons for regional differences in HBV epidemiology in Africa

As exemplified in this study by the case of Uganda, we find evidence of region-specific differences in exposure, transmission, and chronicity rates. This is consistent with patterns previously described in the literature, with the potential for marked differences in seroepidemiology even between neighbouring villages [46]. A diverse range of factors influence the risk of developing chronicity after acute infection (Table 1), with age at exposure among the most robustly recognised. Our data suggest that in regions with low CHB prevalence but high exposure (e.g., central Africa), most exposure events may be occurring in adults. In contrast, in Western Africa, the majority of exposure events may be in early life. Careful data collection and review is required to underpin the most effective interventions for specific locations. HBV genotype may also influence local seroepidemiology, given variations in spatial distributions and genotype-specific epidemiological factors such as variations in transmission potential [47–49]. Likewise, traditional cultural practices that confer exposure to HBV may be common in some regions but not others [50].

**Table 1. Factors that may contribute to regional differences in prevalence of anti-HBc and HBsAg across Africa.**

| Factor | Rationale for contribution to regional differences in HBV seroepidemiology |
|---|---|
| **Circulating HBV viral genotype** | • Predominant genotype varies by region, with genotype-A common in Southern Africa, genotype-D in the North, and genotype-E in the West [51]. |
| **Host ethnicity and genetics** | • HLA-type and T-cell repertoire have been linked to the ability to control the infection [52–54]. |
| **Transmission differences** | • Subtle differences in the transmission patterns (vertical versus horizontal) of the HBV genotypes have been documented.<br>• Transmission route is fundamentally linked to age at exposure [55]. |
| **Age at exposure** | • The probability of developing chronic HBV after exposure is strongly associated with age [56]. Populations with a younger age at exposure are therefore likely to have a higher HBsAg prevalence relative to the anti-HBc prevalence. |
| **Coinfection within population** | • Risk factors for acquisition of blood-borne viruses overlap between HIV, HBV, and HCV.<br>• Egypt and the Nile Delta have some of the highest reported prevalences of HCV globally. Coinfection of HBV and HCV has been linked to spontaneous clearance of HCV although evidence of the impact on HBV remains scarce [57, 58]. |
| **Political instability** | • Central Africa includes several regions disrupted by recent conflict and resulting population migration, with powerful influence on increases in interpersonal violence and sexual assault, reduced access to barrier contraception, inadequate screening of blood products, and reduced access to healthcare, all of which can increase exposure rates in the adult population. |
| **Traditional cultural practices** | • Exposure to blood-borne viruses is influenced by traditional healing practices, scarification, piercing, tattooing, and nonsterile surgical practice. |
| **Uptake of HBV vaccination in the region** | • Countries with earlier uptake of the HBV vaccine are likely to have lower anti-HBc and HBsAg prevalence than countries that implemented the vaccine later.<br>• Prevalence of vaccine escape mutants may contribute, although data for Africa are scarce [45]. |

**Abbreviations**: anti-HBc, antibody to hepatitis B core antigen; HBsAg, hepatitis B surface antigen; HBV, hepatitis B virus; HCV, hepatitis C virus; HIV, human immunodeficiency virus; HLA, human leukocyte antigen

## Relationship between HBV and HIV

We found no evidence that HIV-positive individuals were more likely to be HBV infected or exposed, in keeping with previous reports [49]. This observation reflects different transmission patterns: HIV is less infectious than HBV when transmitted by blood and is largely sexually transmitted in Africa. In contrast, the risk of developing CHB is high in early life and declines with age. However, robust analysis of the influence of HIV on HBV exposure and acquisition is made difficult by limited data. While we were able to identify a large number of HIV-positive cohorts, only 3 of these had directly comparable HIV-negative cohorts. Among all other published cohorts, which we have assumed to be HIV negative, a background prevalence of HIV infection is likely but not clearly reported.

## T&T in the context of HBV guidelines

Current guidelines for the management of CHB recommend treatment only in certain subgroups, based on risk of liver disease [13]. Thus, deployment of a universal T&T campaign, as modelled in this study (see Methods), would treat many individuals who do not meet current criteria. The infrastructure and budgets to support this detailed individual assessment

are not available in many lower- and middle-income country settings, and there is a call for algorithms to be simplified [59, 60]. Relaxing the stringency of assessment for treatment may also have the added benefit of reducing complications from HBV progression, as not all individuals at high risk of cirrhosis or hepatocellular carcinoma (HCC) are detected by current algorithms [61]. Therefore, adopting a universal T&T strategy for HBV would need to be carefully evaluated to consider challenges and risks as well as benefits. Wider deployment of NA therapy involves a need for expanded infrastructure, imposes economic costs, and carries risks of drug side effects and toxicity, as well as raises concerns about nonadherence and the associated potential for increasing the selection of drug resistance [45]. However, these may be offset by the benefits of individual, societal, and economic benefits of reducing the morbidity and mortality of chronic liver disease, as well as the population gains of minimising the risk of transmission.

## Caveats and limitations

Our maps highlight important geographical gaps in HBV data (Fig 3); furthermore, existing cohorts are often relatively small and biased by the recruitment of specific groups. The published literature does not account for the prevalence of occult HBV, which is rarely detected due to lack of availability and high cost of DNA testing. However, individuals with occult HBV would still generate anti-HBc; thus, while we may be underestimating the prevalence of active infection, these individuals are still included within our exposed population.

We considered individuals to have cleared HBV if they were HBsAg negative and anti-HBc positive. However, a proportion of these individuals may still harbour HBV cccDNA in their liver and are at risk of reactivation, particularly in the context of immunosuppression. T&T strategies (whether using HBsAg and/or HBV DNA) would not identify these individuals, and therefore improved diagnostics and monitoring may be of benefit in high-risk populations.

We did not include data for antibody to HBsAg (anti-HBs) prevalence (immunised population) because a limited number of papers report these together with anti-HBc and HBsAg data. The most common reason for study exclusion from the literature review was no anti-HBc prevalence reported (Fig 1). The sensitivity and specificity of serological assays vary [62], which may represent a source of bias or uncertainty in our data set, although our quality appraisal of the literature shows that methods were robustly reported overall. However, making the inclusion criteria more stringent would have limited the number of studies we could include, potentially limiting our overall findings.

Based on the age of adults represented in most of our cohorts and vaccination starting in 1995, we can assume that the majority of participants in the study were unlikely to have been vaccinated in infancy. Future sero-surveys will provide more insights into the impact of routine infant vaccination. We focused on adult populations only, because the age-associated risk of developing chronic HBV is a confounding factor in younger cohorts, making inference about the anti-HBc prevalence challenging across multiple age groups. It would be of interest to determine age-specific prevalence of HBsAg and anti-HBc because age is likely to be an important source of heterogeneity. However, metadata are poorly reported, and we were unable to disaggregate serological data by age.

Our modelled universal T&T intervention does not stratify individuals for therapy but works on the basis of treating anyone who is HBsAg positive. In current clinical practice, guidelines recommend treatment only in the context of high viral load and/or evidence of inflammatory liver disease [13]. However, explicitly stratifying population subgroups for T&T within our framework would have required the inclusion of clinical classes (e.g., liver transaminases, fibroscan scores), which would have added significant uncertainty. Our model

framework does not include explicit age-specific or risk-group assumptions regarding force of transmission, and again we argue that little data exist to inform this parameterization. Despite uncertainty in specific variables, the model fixes certain parameters, based on estimates from the literature (e.g., HBeAg is set at 27%, recognising that the confidence intervals around this point estimate are wide and vary between settings [63, 64]). Our projections are not intended to be exact quantifications of impact over time but serve as a means of comparing the dynamic and nonlinear outcomes of different strategies.

While we have considered HIV as a precedent for T&T interventions, there are also important differences between HIV and HBV, namely the lack of an HIV vaccine and the shorter average timeframes over which morbidity and mortality evolve in the setting of HIV infection.

## Implications for future investigation and practice changes

Our data show the potential value of T&T approaches for HBV, building both on experience gained from HIV and on insights into paediatric interventions that we have previously modelled [9]. Further analysis is required to investigate more specifically the populations or contexts in which vaccination strategies or T&T might best be deployed, and the extent to which this intervention is synergistic with other routine interventions (infant vaccination and PMTCT). We advocate significant investment in capacity building for improving HBV diagnosis and treatment, including point-of-care testing, antenatal screening, and provision of TDF. A sustained and systematic commitment to diagnosis and treatment represents a key component of the journey towards HBV elimination.

## Supporting information

**S1 PRISMA Checklist. The PRISMA Statement for Reporting Systematic Reviews and Meta-Analyses.**
(PDF)

**S1 Table. Population data and HBV seroepidemiology for Uganda used to inform a model to determine impact of interventions.**
(PDF)

**S2 Table. Details of studies from Africa reporting HBV prevalence data from $\geq$2 cohorts.** These studies ($n$ = 13) were each reported in a single publication but contribute $\geq$2 data points to our overall analysis. Differences in the cohorts are highlighted in city/location, cohort characteristics, and cohort size. Studies listed in alphabetical order by location.
(PDF)

**S3 Table. Predicted HBsAg prevalence for Northern, Eastern, Southern, Western, and Central Africa, based on a given anti-HBc prevalence.** WLR was performed using cohort size as weight. Predicted HBsAg prevalence by WLR for Northern, Eastern, Southern, Western, and Central Africa at anti-HBc prevalences ranging from 5% to 95%, increasing in increments of 5%. Values are plotted in S3 Fig. WLR, weighted linear regression.
(PDF)

**S4 Table. Results of a systematic literature review to identify evidence or recommendations for use of catch-up HBV vaccination in adolescents and adults in Africa.**
(PDF)

**S1 Fig. Forest plots of HBsAg and anti-HBc prevalence and proportion of the population remaining susceptible in all studies identified in Africa (1995–2019).** Confidence intervals for the population mean were calculated as $\bar{x} \pm t'^S/_{\sqrt{n}}$, in which S is the standard deviation, $\bar{x}$

the sample mean, $n$ the cohort size, and t' the upper $(1 − C)/2$ critical value for the t distribution with $n − 1$ degrees of freedom. Panels (a) for anti-HBc, (b) for HBsAg, and (c) for susceptible population. Full metadata online at Figshare: 10.6084/m9.figshare.6154598.
(PDF)

**S2 Fig. Average prevalence of anti-HBc and HBsAg in confirmed HIV-positive cohorts and all other cohorts.** Cohort characteristics were recorded for each study (full metadata online at Figshare: 10.6084/m9.figshare.6154598). All cohorts characterised as HIV positive ($n = 27$) were grouped together and compared with cohorts that were not listed as being HIV positive ($n = 79$). Three cohorts testing sex workers ($n = 1$) and patients in HIV testing clinics ($n = 2$) were excluded from the analysis as not all participants were HIV positive in these cohorts, but HIV-positive participants are likely to be enriched in these cohorts. Weighted averages, accounting for study size, are shown along with 95% confidence intervals. No significant differences were identified for either anti-HBc or HBsAg prevalence (both $p = 0.06$ and $p = 0.07$, respectively).
(PDF)

**S3 Fig. Predicted HBsAg prevalence for Northern, Eastern, Southern, Western, and Central regions of Africa with a given total anti-HBc prevalence (reflecting exposure).** WLR was performed using cohort size as weight. Predicted HBsAg prevalence by WLR for Northern, Eastern, Southern, Western, and Central Africa at anti-HBc prevalences ranging from 5% to 95%, increasing in increments of 5%, is presented here. Plotted from values given in S3 Table. WLR, weighted linear regression.
(PDF)

## Author Contributions

**Conceptualization:** Anna L. McNaughton, Janet Seeley, Robert Newton, Ponsiano Ocama, Philippa C. Matthews.

**Data curation:** Anna L. McNaughton, Jolynne Mokaya, Sheila F. Lumley, Donall Forde, Kenneth R. Katumba.

**Formal analysis:** Anna L. McNaughton, José Lourenço, Phillip Armand Bester, Uri Obolski, Tongai G. Maponga, Dominique Goedhals, Sunetra Gupta, Philippa C. Matthews.

**Funding acquisition:** Philippa C. Matthews.

**Methodology:** Anna L. McNaughton, José Lourenço, Phillip Armand Bester, Uri Obolski, Sunetra Gupta, Philippa C. Matthews.

**Project administration:** Philippa C. Matthews.

**Software:** José Lourenço.

**Supervision:** Dominique Goedhals, Sunetra Gupta, Janet Seeley, Philippa C. Matthews.

**Visualization:** Anna L. McNaughton, José Lourenço, Phillip Armand Bester.

**Writing – original draft:** Anna L. McNaughton, José Lourenço, Philippa C. Matthews.

**Writing – review & editing:** Anna L. McNaughton, José Lourenço, Phillip Armand Bester, Jolynne Mokaya, Sheila F. Lumley, Uri Obolski, Donall Forde, Tongai G. Maponga, Kenneth R. Katumba, Dominique Goedhals, Sunetra Gupta, Janet Seeley, Robert Newton, Ponsiano Ocama, Philippa C. Matthews.

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
