## [Decision Letter · Decision Letter 0]

8 Nov 2019

Dear Dr. Matthews,

Thank you very much for submitting your manuscript "HBV seroepidemiology data for Africa provides insights into transmission and prevention" (PMEDICINE-D-19-03078) for consideration at PLOS Medicine. 

Your paper was evaluated by a senior editor and discussed among all the editors here. It was also discussed with an academic editor with relevant expertise, and sent to four independent reviewers, including a statistical reviewer. The reviews are appended at the bottom of this email and any accompanying reviewer attachments can be seen via the link below:

[LINK]

In light of these reviews, I am afraid that we will not be able to accept the manuscript for publication in the journal in its current form, but we would like to consider a revised version that addresses the reviewers' and editors' comments. Obviously we cannot make any decision about publication until we have seen the revised manuscript and your response, and we plan to seek re-review by one or more of the reviewers. 

We expect to receive your revised manuscript by Nov 29 2019 11:59PM. Please email us (plosmedicine@plos.org) if you have any questions or concerns.

We look forward to receiving your revised manuscript. 

Sincerely,

Thomas McBride, PhD

Senior Editor 

PLOS Medicine

plosmedicine.org

1- Please report your SR/MA according to the PRISMA guidelines provided at the EQUATOR site.

http://www.equator-network.org/reporting-guidelines/prisma/

Please provide the completed PRISMA checklist. When completing the checklist, please use section and paragraph numbers, rather than page numbers.

2- Please revise your title according to PLOS Medicine's style. Your title must be nondeclarative and not a question. It should begin with main concept if possible. "Effect of" should be used only if causality can be inferred, i.e., for an RCT. Please place the study design ("A randomized controlled trial," "A retrospective study," "A modelling study," etc.) in the subtitle (ie, after a colon).

3- Please structure your abstract using the PLOS Medicine headings (Background, Methods and Findings, Conclusions).

4- Please remove the third sentence of the Abstract (“HBV has been neglected...”) .

5- In the Abstract Methods and Findings, please include information on the modelled population and timeframe for the estimates, and main outcome measures. For the systematic review portion, please provide the dates of search, data sources, number of studies included, types of study designs included, eligibility criteria, and synthesis/appraisal methods.

6- In the Abstract Methods and Findings, please quantify the main results (with 95% CIs and p values).

7- Line 70 and throughout: Unless there is a statistical reason to do so, please report p values down to p= 0.001 and anything smaller as p<0.001.

8- In the last sentence of the Abstract Methods and Findings section, please describe the main limitation(s) of the study's methodology.

9- At this stage, we ask that you include a short, non-technical Author Summary of your research to make findings accessible to a wide audience that includes both scientists and non-scientists. The Author Summary should immediately follow the Abstract in your revised manuscript. This text is subject to editorial change and should be distinct from the scientific abstract. Please

 see our author guidelines for more information: https://journals.plos.org/plosmedicine/s/revising-your-manuscript#loc-author-summary

10- Please remove the last sentence of the Introduction (“Our results have immediate potential…”).

11- Please include the PRISMA flowchart in the main paper.

12- Please update your literature search to the present time.

13- As noted by reviewer 1, please conduct a meta-analysis.

14- Please evaluate study quality and risk of bias.

15- Though the HBV dynamic model was previously published, please provide more details on the model in the Methods section.

16- Please provide a diagram that shows the model structure, including how the disease natural history is represented, the process and determinants of disease acquisition, and how the putative intervention could affect the system.

17- What were the algorithms used for HBsAg and anti-HBc in the various studies?

18- Does the model take into account non-compliance with drug treatment?

19- Please use the "Vancouver" style for reference formatting, and see our website for other reference guidelines https://journals.plos.org/plosmedicine/s/submission-guidelines#loc-references

Reference 4, for example, is incomplete. 

Comments from the reviewers:

Reviewer #1: Alex McConnachie, Statistical Review

McNaughton et al look at transmission and prevention of HBV in Africa; this review looks at the statistical aspects of the paper.

The paper involves a systematic review, but no meta-analysis. At least, it is not described as such; the analysis of metadata is done with equal weight given to each study, and groups of studies are compared with standard independent samples tests. This seems wrong - studies should be weighted according to their size, or the precision with which they estimate prevalence.

The phrase "non-parametric data" is not correct. It is the test used to analyse the data that is non-parametric, not the data itself. Also, if a non-parametric test is used to compare two groups, then ANOVA for comparing more than two groups is not right; something like a Kruskal-Wallis test would seem appropriate.

The paper then describes the results of adapting a model they have built to explore the impact of adult vaccination, or a Test and Treat approach on HBV control. The model is applied using data obtained from the systematic review specifically from Uganda. The results, shown in Figure 4, appear to include various other strategies, which are not well described in the paper. Also, I do not see the value of projecting the impact of strategies up to 500 years into the future. Even a 50-year horizon is probably beyond most policy-makers considerations.

No attempt appears to have been made to explore the potential impact of different control strategies in different settings. E.g. are there any settings in which a catch-up vaccination strategy would be effective, or in which a T+T strategy less so? There is some speculation about this in the discussion, but it would be better to use the model to investigate what the optimal strategies might be in different settings.

Reviewer #3: This manuscript describes a systematic review on the prevalence of chronic HBV and modelled estimates of the impact of catch up vaccination or test and treat in Africa. The results suggest heterogenous prevalence of chronic HBV with regional variation. Catch-up vaccination has predictably little impact on chronic HBV prevalence. Test & Treat appears to be an effective method for reducing the prevalence of chronic infection albeit that the impact will take some time to achieve SDG goals. 

The wording of the abstract is a little confusing - "relationship between the prevalence of HBsAg and anti-HBc (p<0.0001)" All patients who are HBsAg positive are also anti-HBc positive. It would help if the authors made it clearer that they are talking about prevalence of chronic infection and previous acute infection (exposure)

The term 'infection' is insufficiently precise. It is important to distinguish acute HBV infection from chronic HBV infection as the risk of cirrhosis and hepatocellular carcinoma are only associated with the latter

Whilst the authors are correct in asserting that catch-up vaccination in adolescents is still being proposed as a public health strategy, in reality this is now rarely advocated. It is well understood that 1. Exposure to the infection takes place earlier in life and 2. Exposure as an adolescent rarely results in chronic infection and is therefore a much less important public health issue. The only situation where catch up vaccination is potentially beneficial is in regions where HDV co-infection is prevalent and spread through sexual transmission

The HBeAg positivity rate amongst those who are HBsAg positive is high (Suppl Table 1) . The rate will vary according to age and is most relevant in pregnant women

Rate of HBeAg clearance uses an estimate from Taiwan rather than data from West Africa - see Shimakawa et al 

The paper does not seem to take into account current indications for treatment. If advocating treatment for all patients with chronic infection irrespective of disease phase, level of viraemia etc then this needs discussing in more detail. 

Whilst the discussion encompasses consideration of exposure and chronic infection rate variation between African regions it missed the wide variation in prevalence and exposure at a much more granular level - see Whittle JID 1990

I am not comfortable with postulating a role for insect vectors in HBV. This has been examined extensively and is not supported by evidence. 

Reviewer #4: Dear authors,

This is a well-written, excellent, comprehensive and creative article. The authors start with and support the hypothesis that testing and treating hepatitis B may be a viable public health option. The comprehensive literature review of hepatitis B serology and the modelling and compilation of data are clear and well-presented. My comments concern the paper itself.

There are three points that could be better introduced and discussed in the conclusions and recommendations, so that the appropriate focus is brought to the findings. An alternate presentation of data should be considered.

Firstly, the authors state at the outset that there are no recommendations for wide-scale catch-up immunization of adults. Yet the scenario of a single one-off all ages catch-up vaccination campaign is one of the interventions modelled and studied in this paper. The authors then conclude that 'Although it can seem intuitive to deploy catch-up vaccination for adults in high prevalence settings... only a minority will benefit from the intervention'. Therefore they appear to advocate a blanket dismissal of 'catch-up' vaccination which they state is not actually the subject of any recommendation. However the authors have not analyzed targeted vaccination strategies such as for example catch-up vaccination in low EPI coverage settings for young children to reduce childhood horizontal transmission, or assessed alternate booster/catch-up approaches in school-aged children prior to sexual début. So the emphasis on a non-recommended strategy risks detracting from consideration of possibly more useful vaccination and other alternatives. It would be far preferable for example to have the abstract, intro and discussion focus on the impact of vaccination at birth and/or after school entry as these are also modelled and far more relevant from a public health programmes perspective.

The presentation of the models out to 500 years is nice but fanciful, and detracts from closer assessment of alternate strategies, and possible combinations of strategies, over the next ten to 50 years. 

My recommendation is to refocus the abstract, introduction and discussion on the range of possible public health interventions that should be considered, not on the one for which they found no recommendations.

The authors should consider also presenting findings to at most 50 or 100 years (or event 200, the relevance of which is also highly questionable) so that readers can really focus on the immediate public health implications of the paper. Highlighting the 2030 goal would add immediacy to the findings. I also recommend removing the data points for the one-off all ages vaccination campaign, as it also detracts from the far more relevant findings. Mentioning those results briefly in the text would be sufficient.

Secondly, the primary thesis of the paper concerns treatment of hepatitis B, yet there is no mention of treatment options upon which the paper relies. Please include a paragraph outlining the treatment options currently being used and some parameters considered (e.g. assumptions about what constitutes successful treatment or assumed success rates, mention of complications attributable to treatment). This does not have to be detailed, but sufficient to orient the reader to the public health intervention being proposed and argued for in this paper.

Thirdly, the practical, logistical and economic feasibility of the T&T intervention is likely at this time to be quite low in the African continent and countries included in the study. The authors draw an analogy with HIV T&T, but there is no HIV vaccine, and those infected remain infectious and face morbidity and mortality much more quickly than chronic HPV carriers do, so it may be an inadequate comparison. The cost is likely to be prohibitive and the systems not in place. Perhaps cervical cancer screening would be a reasonable analogy. In order to orient the reader and provide more targeted recommendations, it would be helpful to provide any examples of where the T&T approach is being piloted for HBV and reference practical considerations. 

From a public health programming perspective and the 2030 targets, the most relevant question would be what would be the success of a primary prevention strategy combining 1.PMTCT 2.Routine neonatal HB vaccination and 3.School-based booster/catch-up HB vaccination. Then the question could be asked what is the marginal added value of a T&T intervention at scale. If this paper cannot address such questions now, then it would be helpful to set up the discussion to indicate what are the appropriate questions for public health programmes in Africa, so that subsequent research can pursue them.

Thanks very much for giving me the opportunity to review your excellent paper and I look forward to your further work.

[LINK]

---

## [Decision Letter · Decision Letter 1]

30 Jan 2020

Dear Dr. Matthews,

Thank you very much for re-submitting your manuscript "Hepatitis B virus seroepidemiology data for Africa: a systematic review and meta-analysis to inform analysis of optimal local intervention strategies" (PMEDICINE-D-19-03078R1) for consideration at PLOS Medicine.

I have discussed the paper with editorial colleagues and our academic editor, and it was also seen again by the reviewers. I am pleased to tell you that, provided the remaining editorial and production issues are dealt with, we expect to be able to accept the paper for publication in the journal.

[LINK]

Please let me know if you have any questions. Otherwise, we look forward to receiving the revised manuscript shortly. 

Sincerely,

Richard Turner PhD, for Thomas McBride, PhD

rturner@plos.org

Requests from Editors:

Please confirm that all other authors have agreed to the inclusion of a new author (UO). 

We ask you to amend the title so that it better matches journal style, and suggest "Analysis of hepatitis B virus seroepidemiology data for Africa and modelling of intervention strategies: a systematic review and meta-analysis".

In your abstract, around line 71, we suggest adding a sentence, say, to quote aggregate data from the studies found (e.g. those noted at lines 286-287). 

We also suggest adding some quantitative details of the "region specific differences". 

In the abstract and main text, you quote an estimate of "33%" for reduction of HBV incidence at 20 years with test and treat programmes, for example. We notice a shaded area in Fig 6E, for example, and wonder whether this refers to a credibility interval, say (we may have missed an explanation in the text)? We ask you to add indications of uncertainty to this and other estimates in your paper, if available. 

At line 115, please substitute "existing" for "(previously published)". Also, please soften the wording here to reflect that modelling is involved, e.g., substituting "... is predicted to have a substantial impact" for "has". 

Early in the methods section, please state whether the study had a protocol or prespecified analysis plan, and if so attach the relevant document(s) as a supplementary file. Please highlight analyses that were not prespecified.

Please restructure the early part of the discussion section of your main text so that the first paragraph provides a summary of the study's findings.

At line 493, please make that "report data ... these" (rather than "this").

Please make that "potential value" at line 533.

Throughout the text, please format reference call-outs as follows: "... 2030 targets [9,10].".

Please abbreviate journal names consistently through the reference list.

For reference 1, please correct the group author name and add full access details. 

We suggest removing reference 29, and including the access details at the start of your "Results" section. 

For reference 31, "S Afr Med J" will suffice as the journal name. 

Please update reference 55. 

You mention a completed PRISMA checklist, but we did not find this among the submitted files. Please include this as a separate supplementary file with your revision; and ensure that individual items in the checklist are referred to by section (e.g., "Methods") and paragraph number rather than by page or line numbers, as the latter generally change upon publication. Please refer to the checklist in the main text (methods section). 

Comments from Reviewers:

*** Reviewer #1: 

Alex McConnachie, Statistical Review

I thank the authors for their responses to my original points. These are satisfactory, and I have no further comments to make.

One very minor thing I missed first time around. Line 335 includes "p>0.05" - it is usually preferable to report actual p-values.

*** Reviewer #2: 

has addressed all comments 

*** Reviewer #3: 

I am happy with the authors responses and the revised version of the manuscript. I would make one minor recommendation - It is not entirely true to say that Test & Treat has not been evaluated, as suggested in the abstract. The strategy has been piloted in The Gambia as detailed in references 19 & 44

*** Reviewer #4: 

In this revised manuscript, the authors have met most of the recommendations of the previous review, including my own, resulting in clearer and more relevant presentation and more nuanced discussion and reommendations.

In referring to outcome of models, it would be preferable to use terms such as 'our models suggest that...' rather than 'we have demonstrated that', and 'would' or 'could' rather than definitive statements such as 'was' or 'will', including in the plain language summary.

***

[LINK]

---

## [Editor Report · Decision Letter 2]

13 Mar 2020

Dear Prof Matthews, 

On behalf of my colleagues and the academic editor, Dr. Mirjam Kretzschmar, I am delighted to inform you that your manuscript entitled "Hepatitis B virus seroepidemiology data for Africa: modelling intervention strategies based on a systematic review and meta-analysis" (PMEDICINE-D-19-03078R2) has been accepted for publication in PLOS Medicine. 

PRODUCTION PROCESS

PRESS

PROFILE INFORMATION

Thank you again for submitting the manuscript to PLOS Medicine. We look forward to publishing it. 

Best wishes, 

Richard Turner, PhD

Senior Editor 

PLOS Medicine

plosmedicine.org